# How Many Azores Bullfinches *(Pyrrhula murina)* Are There in the World? Case Study of a Threatened Species

Tarso de M. M. Costa [1,2,*] , Artur Gil [3,4] , Sergio Timóteo [5] , Ricardo S. Ceia [6,7,8] , Rúben Coelho [1,9] and Azucena de la Cruz Martin [1]

1   Sociedade Portuguesa para o Estudo das Aves, Rua António Alves de Oliveira, 9630-147 Nordeste, Portugal; ruben.ml.coelho@azores.gov.pt (R.C.); azucena.martin@spea.pt (A.d.l.C.M.)
2   cE3c—Centre for Ecology, Evolution and Environmental Changes & ABG—Azorean Biodiversity Group, Faculty of Sciences and Technology, University of the Azores, 9500-321 Ponta Delgada, Portugal
3   IVAR—Research Institute for Volcanology and Risks Assessment, University of the Azores, 9500-321 Ponta Delgada, Portugal; artur.jf.gil@uac.pt
4   cE3c—Centre for Ecology, Evolution and Environmental Changes & ABG—Azorean Biodiversity Group, CHANGE—Global Change and Sustainability Institute, Faculty of Sciences and Technology, University of the Azores, 9500-321 Ponta Delgada, Portugal
5   Department of Life Sciences, Associate Laboratory TERRA, Centre for Functional Ecology, University of Coimbra, 3000-456 Coimbra, Portugal; sergio.timoteo@uc.pt
6   CIBIO/InBIO, Centro de Investigação em Biodiversidade e Recursos Genéticos, Universidade do Porto, 4485-661 Vairão, Portugal; ricardoceia@gmail.com
7   CIBIO/InBIO, Centro de Investigação em Biodiversidade e Recursos Genéticos, Instituto Superior de Agronomia, Universidade de Lisboa, 1349-017 Lisboa, Portugal
8   BIOPOLIS Program in Genomics, Biodiversity and Land Planning, Universidade do Porto, 4485-661 Vairão, Portugal
9   Secretaria Regional do Ambiente e Alterações Climáticas, Rua do Galo, nº 118, 9700-091 Angra do Heroísmo, Portugal
*   Correspondence: tarsommc@gmail.com

**Abstract:** The Azores bullfinch *(Pyrrhula murina* Godman, 1866) is a rare Passeriformes endemic from the eastern part of São Miguel Island, Azores, Portugal. This bird was almost considered extinct in the first half of the 20th century, but due to recent conservation measures, it has experienced a recovery since the beginning of the 2000s. Despite the attention given to this bird, the size of its population is still controversial, and the most recent studies present significant divergences on this behalf. The purpose of the present study is to present data from the long-term monitoring and results of the third single-morning survey of the Azores bullfinch to update information about the population size and range of this species. In addition, we performed a literature review to highlight the limitations and advantages of the different approaches for monitoring this species. The Azores Bullfinch records during the single-morning survey indicated a reduction in the extent of occurrence and area of occupancy of this species in comparison with the previous studies, despite the increase in bird detection. However, we suggest that the distribution range of this species needs further analysis concerning its area to exclude non suitable habitats from this analysis. In this study, we conclude that the most likely size of the Azores bullfinch population is 500 to 800 couples, with a slow population growth tendency and an area of distribution of 136.5 km$^2$.

**Keywords:** bird census; conservation; *Pyrrhula murina*; oceanic islands; Macaronesia; Laurel Forest

## 1. Introduction

The Azores Bullfinch *(Pyrrhula murina* Godman, 1866) is the unique endemic Passeriformes species in the Azores Archipelago (Portugal) and solely inhabits the eastern part of São Miguel Island [1]. The life cycle of this species is intrinsically related to the Azorean Laurel Forest [2,3], one of the most threatened ecosystems of the Macaronesian biogeographic region [4]. Due to human-induced habitat modification and degradation, only

3% of its original 200,000 ha. distribution remains nowadays [5]. In addition to further habitat destruction [6] and climate change [7], the current main threat to Laurel Forest, especially in the Azores, is invasive species that outcompete native Laurel Forest plant species, completely transforming the habitat and thus leading to the extinction of endemic species of different taxa [8–10].

Although the Azores bullfinch was probably abundant in the second half of the 19th century [11], it is currently one of the rarest bird species in Europe and the one with the smallest distribution area [1]. The degradation of its original habitat forced the Azores Bullfinch to search for alternative food sources. Being a bird that feeds mainly on flowers, fruits, seeds, and ferns [12,13], orange blossoms became an essential resource for its survival during the 19th century [1]. The Azores bullfinch was then considered a plague for orange production, which represented an important economic activity in São Miguel Island during this period, and for this reason, it was hunted by locals. Likely because of this hunting pressure, many naturalists who visited São Miguel Island in the first half of the 20th century noticed that the Azores' bullfinch had become very scarce [14].

The first actual estimates of the Azores bullfinch population were obtained in the second half of the 20th century and showed a low number of individuals. At the end of the 1970s, it was estimated that only 30 to 40 couples existed [15], and at the end of the 1980s, the Azores bullfinch population was estimated to be around 100 birds [16,17]. Due to the low number of individuals and its reduced geographic range, the Azores bullfinch's conservation status was considered "Critically threatened of extinction", according to the IUCN Red List [18].

To better understand this species' biology and ecology (namely population size and distribution), the Azores bullfinch has been monitored annually since 1991 (except for the period between 1997 and 2001). This monitoring consisted of point counts distributed throughout the distribution area of the species. The analysis of the data from 1991 to 2008 showed that the Azores bullfinch experienced a significant population increase at the beginning of the 2000s [19]. It further showed that between 2005 and 2006, the number of individuals doubled. This increase in population size resulted in an update to its conservation status. In 2010, the Azores bullfinch's status was downgraded to "Threatened", and, in 2016, it was reclassified as "Vulnerable", which represents the lowest category for species at risk of extinction [18]. Such improvements likely result from several actions dedicated to its conservation, namely habitat restoration efforts and the implementation of scientific works regarding its ecology [14].

The work developed throughout the Ph.D. thesis of Ramos [1] and the discussion of the first action plan for Azores bullfinch conservation [20] was crucial for implementing appropriate measures for this bird's protection. The main threats identified for the Azores bullfinch were habitat loss and low food availability, especially during the winter [13], as an indirect effect of the degradation of their habitat by the expansion of exotic invasive flora and the consequent reduction of food sources. Therefore, priority conservation actions were to control the spread of the invasive flora, responsible for much of the Laurel Forest devastation, along with the recovery of the habitat quality to increase the availability of feeding resources provided by the native flora species on which this bird mostly forages [21]. Permanent monitoring was also considered a high priority and involved the annual estimation of the population size and breeding success. The most recent studies on this rare species addressed its ecological distribution and relation to natural and exotic vegetation [3], population dynamics [22], status assessment [19], phylogeny [23], and the impacts of introduced rodent predation [24].

With three different conservation projects funded by the LIFE Programme of the European Commission, the Portuguese Society for the Study of Birds (SPEA) has ensured annual monitoring of the Azores bullfinch since 2003. This yearly monitoring allows for estimating the population size and trends of the Azores bullfinch. In addition, some observations regarding the bio-ecology of this bird, such as reproductive success, feeding patterns, and distribution, are registered during the annual monitoring.

The present study analyzes the data obtained from the last 12 years of Azores bullfinch monitoring to update information about its population size and the conservation status of this endangered species. In addition, the previous literature is reviewed to highlight the limitations and advantages of the different approaches and propose a way forward to better understand the Azores bullfinch population and its conservation.

## 2. Materials and Methods

### 2.1. The Azores Bullfinch Distribution Area

The Azores bullfinch distribution area (Figure 1) comprises the mountainous region in the eastern part of São Miguel Island (Archipelago of the Azores, Portugal), included in two Natura 2000 sites, namely the Special Protection Area (SPA) "Pico da Vara/Ribeira do Guilherme", and the Site of Community Importance (SCI) "Serra da Tronqueira/Planalto dos Graminhais". The habitat of the Azores bullfinch is the Azorean Laurel Forest; however, due to human intervention, it has been substantially altered. For this reason, the distribution area of the bird includes spots of Laurel Forest (with different degrees of degradation due to invasive alien plant species presence) and anthropogenic landscapes such as pastures, roads, and agriculture fields.

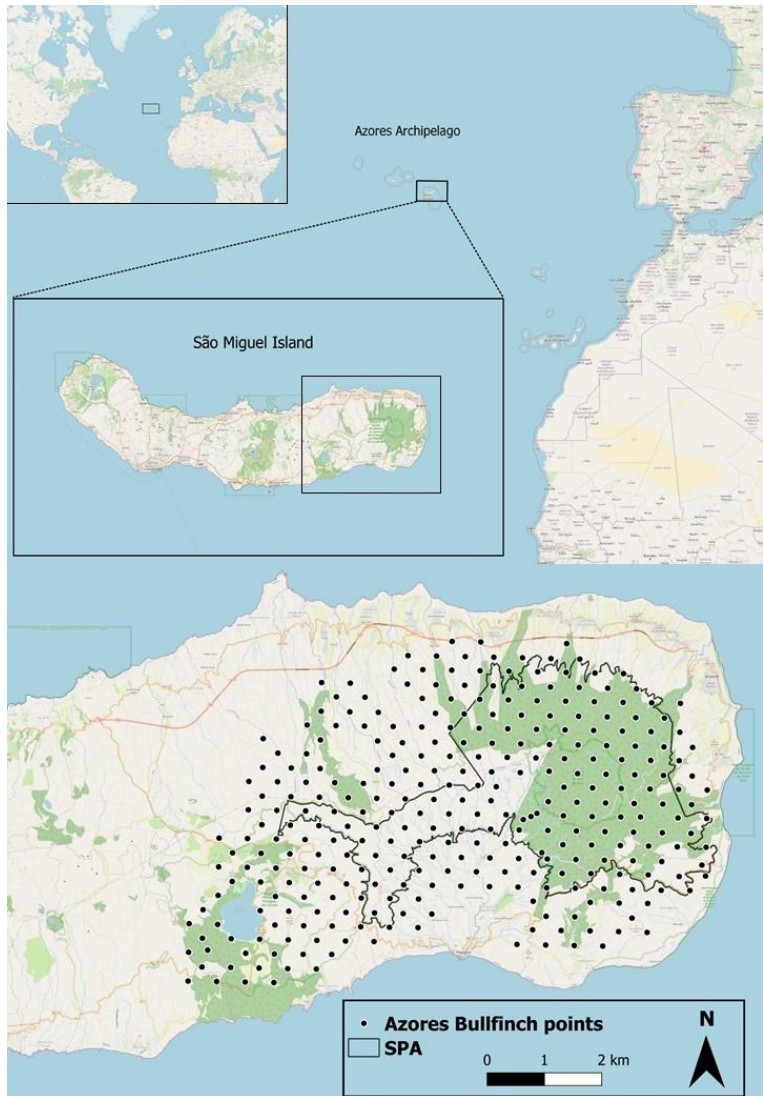

**Figure 1.** Location of the Azores bullfinch distribution area and the counting points used in the single-morning study. SPA—Special Protected Area Pico da Vara/Ribeira do Guilherme.

### 2.2. Azores Bullfinch Monitoring

The data collection took place throughout the Azores bullfinch distribution area from 2009 to 2021. Fieldwork was performed in June and July during the reproduction period of the species to increase the detectability of the birds [25], on days with favorable climatic conditions (absence of rain, fog, and strong winds) from 6:30 h to 11:30 h AM (GMT-1). Previous training exercises were provided to the observers in order to make them able to perform the census. Training included exercises to accurately estimate distances for bird detection.

Monitoring consisted of two different methodological approaches: a single morning survey and annual monitoring. The single morning survey was employed in 2016, aiming to obtain more accurate distribution and population size data. The single morning survey involved 60 volunteers distributed across 307 counting points (including the 158 counting points from the annual monitoring plus 148 additional new counting points), covering an area of nearly 15,200 hectares (Figure 1). 2016 was the third edition of the single morning survey, which was previously implemented in 2008 and 2012 [19,26]. The annual monitoring consisted of registering all birds heard or seen for 8 min across 158 counting points covering an area of nearly 9000 hectares. One or two observers conducted the point counts for approximately one month. In both methodologies, the counting points were established at the corners and centers of the squares of the 1 × 1 km units UTM grid superimposed in the study area. The annual monitoring was not performed in the years when the single-morning study was conducted.

For the bird counting during the single morning survey, the variable circular plot method (VCPM; [27]). All *P. murina* individuals seen or heard were recorded following the recommendations of [28] for snapshot surveys. To do this, we adopted the methodology of the previous Azores bullfinch studies performed by [19,26] that defined the "snapshot" moment as precisely five minutes after arrival at the point.

### 2.3. Data Analysis

#### 2.3.1. Range Size

Using the data from the single morning survey, we estimated the range of the Azores bullfinch by using the occurrence and occupancy areas (IUCN, 2001). The occurrence area corresponds to the minimal polygon that includes the UTM 1 × 1 km units with bird detections. The occupancy area is the sum of the UTM 1 × 1 km units with bird detections.

#### 2.3.2. Population Trend

The annual population index is the estimation of the number of birds at the counting points during the annual monitoring of the Azores bullfinch. This index was employed before to describe the long-term trend in the Azores bullfinch population size [19]. The population index is obtained through a Generalized Linear Mixed Model (GLMM). The response variable was the count of birds at an individual point, assuming a Poisson distribution and log-link function. The coefficient estimates for each significant variable were back-transformed, producing an annual population index using the following equation:

$$\exp^{(\text{intersection coefficient + previous year's estimate})}$$

The population index was calculated from 2008 to 2021 (except in the years of the single-morning study, namely 2008, 2012, and 2016), providing a long-term trend.

#### 2.3.3. Population Density and Size

The population density was estimated using the data from the single morning surveys performed in 2012 and 2016. The data acquired during the 8-min and snapshot surveys was analyzed to calculate densities (individuals $\times$ ha$^{-1}$) using the software DISTANCE version 7.3 [29]. The analysis of distance sampling data consisted of three steps: exploratory analysis, model fitting and selection, and model inference. Available key functions (uniform,

half-normal, hazard rate, and negative exponential) were tested with cosine series expansion. The selection of a suitable detection function was guided by Akaike's Information Criterion (AIC), corrected for small sample size [30,31], v2 model-fit statistics, and visual inspection of detection probability and probability density plots [28]. The population size for 2016 was estimated by multiplying the computed density by the total sampled area (15,200 ha).

## 3. Results

### 3.1. Azores Bullfinch Detection

During the annual monitoring from 2009 to 2021, the Azores bullfinch detections ranged from 70 to 119, with an average of 95 detections (±14.6 standard deviation). Figure 2 summarizes all the Azores bullfinch detections achieved only in the annual monitoring surveys from 2009 to 2021 (excluding the Priolo Atlas 2012 and 2016).

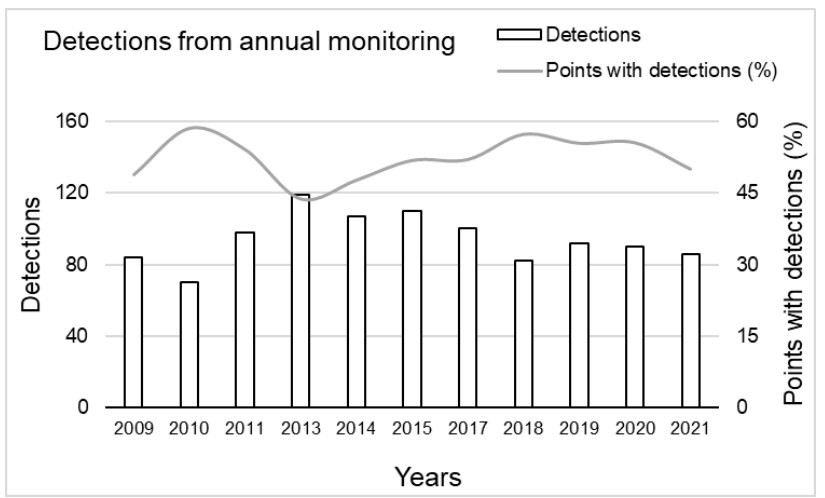

**Figure 2.** Summary of Azores bullfinch detections in the annual monitoring, from 2009 to 2021. The secondary axis shows points with detection percentages.

The observed distribution from a single morning survey (Figure 3) and annual monitoring (Figure 4) suggest the existence of three hot spots in the Azores bullfinches along the ZPE Pico da Vara Ribeira do Guilherme, namely in the north (near Algarvia), in the southeast (near the Serra da Tronqueira), and in the west (near the Graminhais plateau and Povoação).

### 3.2. Range Size

The Azores bullfinch distribution of the records of the single morning survey from 2016 resulted in an extent of occurrence of 130 km$^2$ and an area of occupancy of 52 km$^2$ (Figure 3).

Taking into account all the annual surveys performed from 2009 to 2021, including the single morning surveys done in 2012 and 2016 (Priolo Atlas editions), all the Azores bullfinch records resulted in an overall extent of occurrence of 136.5 km$^2$ and an area of occupancy of 93 km$^2$ (Figure 4).

### 3.3. Population Size

The snapshot count at the fifth minute resulted in 23 Azores bullfinch records in the single-morning study of 2016. Therefore, a density of 0.08 birds × ha$^{-1}$ was estimated through the key function Half Normal, with a confidence interval (95%) of 0.04 and 0.16 lower and upper limits, respectively. The estimated population size (bird density multiplied by the size of the study area) is 1216 individuals (a minimum of 608 and a maximum of 2432 birds).

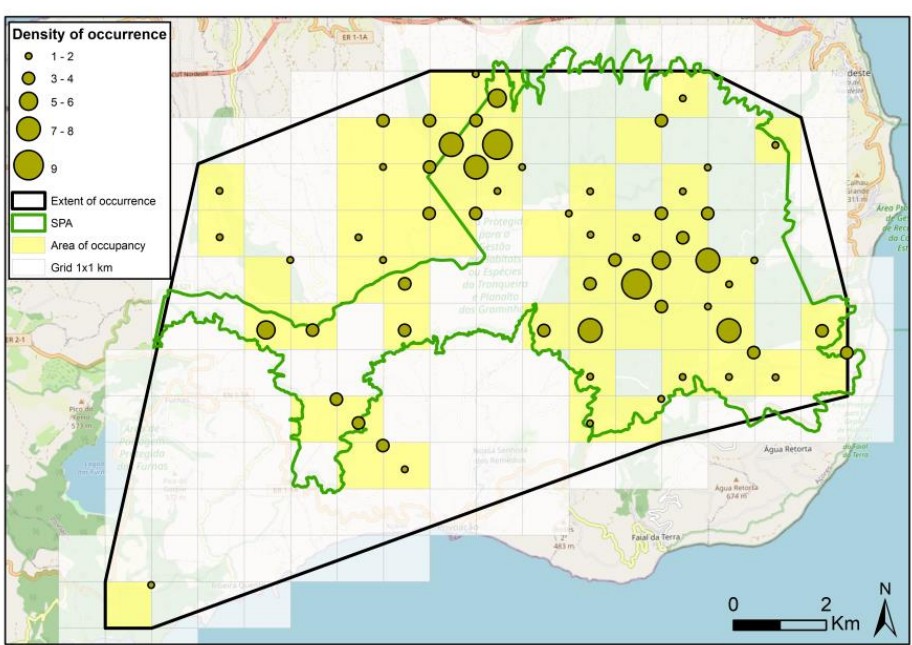

**Figure 3.** Density map illustrating the number of Azores bullfinch observations in every counting point used in the 2016 single morning survey (Priolo Atlas 2016).

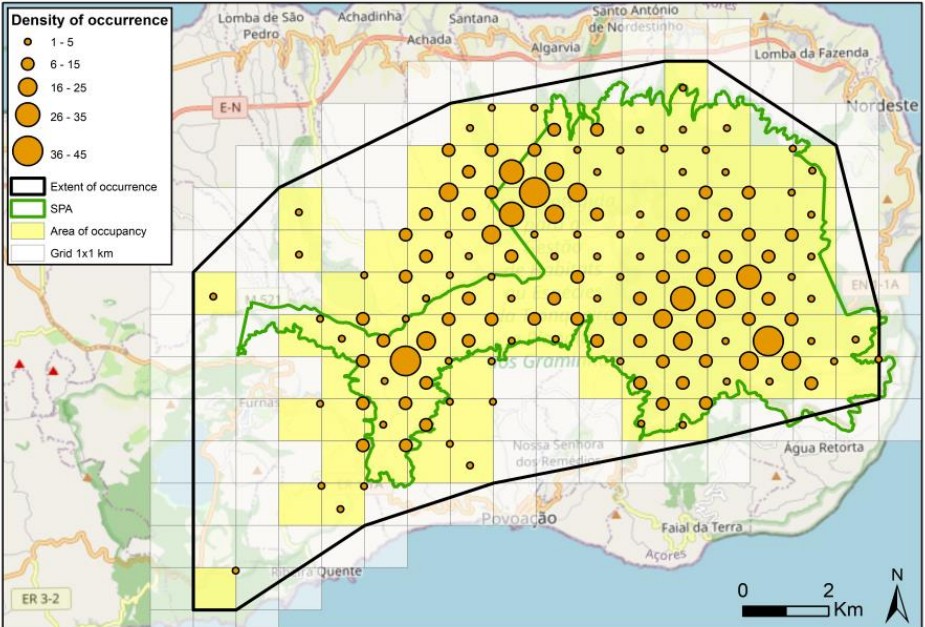

**Figure 4.** Density map illustrating the number of Azores bullfinch observations in every counting point used in the annual surveys, between 2009 and 2021 (including the Priolo Atlas 2012 and 2016).

*3.4. Population Trend*

The population index ranged from 0.44 (2019) to 0.75 (2016), with an average of 0.57. In general, between 2008 and 2021, the population index showed a slight oscillation, with values similar to the series average despite the large width of the 95% confidence intervals (Figure 5).

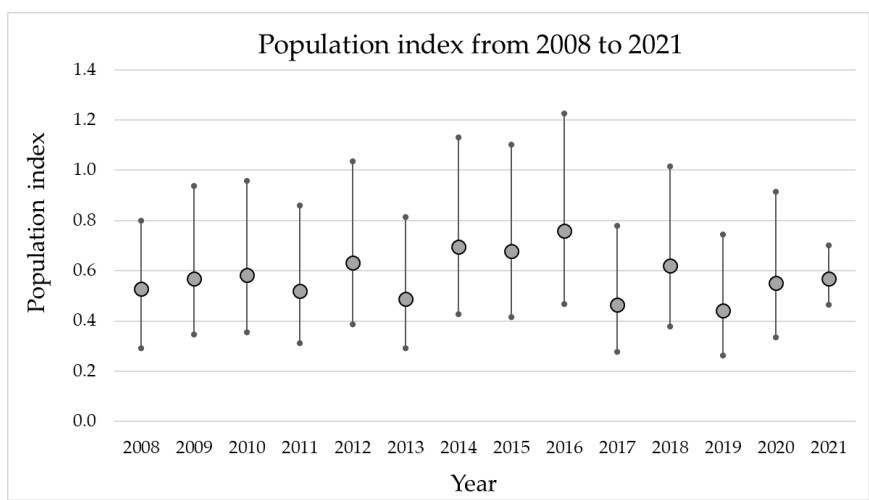

**Figure 5.** Azores bullfinch population trend estimated through GLM from the number of birds detected from 2008 to 2021. Vertical bars denote 95% confidence intervals.

## 4. Discussion

The Azores bullfinch records during the single morning survey in this study indicated a reduction in the extent of occurrence and area of occupancy of this species in comparison with the previous studies. Furthermore, results obtained in 2008, 2012, and 2016 single-morning studies suggest that the extent of occurrence and area of occupancy decreased from 2008 to 2016. On the other hand, despite this apparent reduction in the geographic range of the Azores bullfinch, the number of birds registered and the percentage of points with detections in this study were higher in comparison with those observed in the previous single-morning studies. Table 1 summarizes the results obtained in the single-morning studies from 2008, 2012, and 2016 regarding geographical range and bird detection. The controversial results between the apparent reduction of the extent of occurrence and area of occupancy and the increase in bird detection suggest that the pattern of distribution can be changed from 2008 to 2016. Despite the Azores bullfinch's ability to nest outside its natural habitat, using exotic trees to install and build its nest [32], it has been noticed that the Azores bullfinch is still very dependent on the native Laurel Forest for food and shelter. In these areas, their abundance is significantly higher than in other habitats across the species distribution area [3]. Therefore, for a broader understanding of the pattern observed in the single-morning studies, it is necessary to perform analysis in the distribution area in order to map the Azores bullfinch habitats and provide information regarding their level of quality in terms of food sources. According to [21], the pattern of distribution of the Azores bullfinch is shaped by the spatial and temporal availability of food sources. For example, due to the location of food sources, from May to December, the Azores bullfinch shows a preference for the edge of the Laurel Forest, while from January and April, it inhabits mainly inside this forest [21]. Considering that the diet of the Azores bullfinch is composed mostly of flowers, fruits, seeds, and ferns, the change in vegetation cover inside their distribution area can lead to changes in their distribution pattern independently of an eventual variation in population size.

**Table 1.** The number of birds detected and range size estimations during single morning surveys for Azores bullfinch population assessment.

| Year | Birds Registered | Points with Detections (%) | Extent of Occurrence (Km²) | Area of Occupancy (Km²) | Study |
|------|------------------|----------------------------|----------------------------|-------------------------|-------|
| 2008 | 90 | 15.6 | 144 | 83 | [19,26] |
| 2012 | 87 | 16.4 | 137 | 90 | [26] |
| 2016 | 99 | 19.8 | 130 | 52 | This study |

The detections of Azores bullfinch in the single-morning surveys from 2008, 2012, and 2016 presented significant differences in terms of distribution. The Azores bullfinch detections in 2008 and 2012 presented a larger cover in the distribution area of this species than in 2016. In 2008 and 2012, the detection was spread both in the center and in the limits of the study area, while detections in 2016 were more concentrated near the center. This distribution pattern resulted in the observed reduction in the extent of occurrence observed in this study. Comparing the location of detections from the single-morning surveys in 2008, 2012 [26], and 2016 (this study) with the cumulative detections between 2009 and 2021, shown in Figure 3, it is possible to note that one of the main hot spots of Azores bullfinch distribution, that is, the west part of the study area, presented few detections in the single-morning survey of 2016. The data in this study, therefore, resulted in a higher number of bird detections but were grouped in fewer areas in comparison to the previous studies. Consequently, the area of occupancy of the Azores bullfinch presented a reduction in 2016 when compared with 2008 and 2012. However, the methodology of the single-morning survey is strengthened to provide an instantaneous picture of the Azores bullfinch population and to reduce re-counting, but it has constraints related to not extending on time. This means that any factor responsible for decreasing detectability, such as human activity around, weather conditions (that can vary significantly on the local scale because of the existence of several microclimates due to the Azores geography), or even the absence or lower activity of birds in a given area on the day of the census, cannot be attenuated by the impossibility of revisiting the area again. Therefore, the reduction of the extent of occurrence and area of occupancy observed in this study needs further data collection in order to be better understood.

The population size of the Azores bullfinch seems to be a controversial issue when we compare the data from this study with estimations provided in the literature (Table 2). There is an apparent disagreement between the study of [26], which estimated around 3500 individuals during the single morning surveys from 2008 and 2012, and previous studies estimates of 1064 [19] and 1608 [22] individuals.

**Table 2.** Comparison of density estimates from single morning surveys for 2008, 2012, and 2016. Data from the present study and those acquired from the literature were obtained through snapshot methodology and analyzed with DISTANCE software. CV—Coefficient of variation; CI—Confidence interval; LL—Lower limit; UL—Upper limit.

| Year | Number of Observations | Density (birds × h$^{-1}$) | Key Function | %CV | 95% CI LL | 95% CI UL | P ($X^2$) | Study |
|------|------|------|------|------|------|------|------|------|
| 2008 | 22 | 0.07 | Uniform | 28.05 | 0.04 | 0.12 | 0.05 | [22] |
| 2008 | 25 | 0.22 | Harzard-rate | 38.51 | 0.11 | 0.46 | 0.20 | [26] |
| 2012 | 29 | 0.26 | Harzard-rate | 37.54 | 0.12 | 0.52 | 0.20 | [26] |
| 2016 | 23 | 0.08 | Half-normal | 31.26 | 0.04 | 0.16 | 0.79 | This study |

Except for [22], all population size estimations were built from a single morning survey also used for the present study. This methodology is particularly interesting since the observers are distributed simultaneously at the counting stations, reducing substantially the risk of counting the same bird more than once. Furthermore, this methodology produces a snapshot of the population, providing valuable data to understand this species distribution pattern. However, the demand for many observers (more than 50) potentially increases the errors in the distance estimation of detections, which is one of the four critical assumptions of distance sampling methodology [32]. However, extensive training was previously provided for the observers, not only to make them able to identify the Azores bullfinch visually and auditorily but also to accurately estimate its distance. In this study (and in the previous single-morning studies as well), all volunteer observers roamed five stations at which objects were previously positioned at known distances. In each station, the volunteers (in groups) estimated the distance of five objects. Distances were revealed at the end of each exercise, allowing observers to adjust their predictions to the real distance of objects.

At the end of the distance training, the distance estimations of each volunteer presented decreasing errors from the first to the last visited station. Therefore, the errors associated with distance estimations could be minimized despite the high number of observers using this methodology. Moreover, even assuming that a higher number of observers does not produce random errors in bird distance estimation, the number of detections obtained from the snapshot count was much lower than the 60 observations recommended to accurately estimate density in distance sampling [28]. Table 2 presents the number of observations and model parameters from the present study and other works in which the population size of the Azores bullfinch was estimated from single morning surveys using snapshot counts. Note that, despite the low number of observations, all models could be validated by the goodness-of-fit test ($\chi$2), and the coefficients of variation are acceptable. There are relevant differences between some parameters presented in Table 2 regarding the density estimate, which shows two different scenarios: the first one presented by [26] and the second suggested in this study and by [19]. The dissimilarity between these two scenarios is well represented in the population size calculated (Table 3). Whether we choose one or the other, we would assume that the Azores bullfinch population is around 3500 or 1000 individuals.

**Table 3.** Population size estimates from different studies. n—Population size; Min.—Minimum population size; Max.—Maximum population size.

| Population Size | | | Sampling Period | Method | Study |
|---|---|---|---|---|---|
| Min. | n | Max. | | | |
| 1282 | 1608 | 1934 | 2005–2008 | Mark-recapture | [22] |
| 608 | 1064 | 1824 | 2008 | distance sampling approach | [19] |
| 1672 | 3344 | 6992 | 2008 | distance sampling approach | [26] |
| 1824 | 3952 | 7904 | 2012 | distance sampling approach | [26] |
| 608 | 1216 | 2432 | 2016 | distance sampling approach | This study |

To determine the most probable size of the Azores bullfinch population and, thus, to evaluate its degree of threat more appropriately, the data provided by distance sampling models need validation. Beyond the uncertainty of the data published by different studies, we must assume the abovementioned constraints of this methodology. In this sense, the study carried out by [22] seems the most appropriate to be used to validate the models produced by the distance sampling method. Ref. [22] used the mark-recapture method by ringing Azores bullfinch individuals, which seems the most robust approach. This study used data from 11 ringing stations covering the four main Azores bullfinch habitats during 25 months (from 2005 to 2008), between May and October, thus reducing the effects of seasonality. The authors estimated an Azores bullfinch population of 1608 individuals by estimating catch probabilities. Since the Azores bullfinch abundance is strongly influenced by seasonality and habitat typology in their distribution area (Ramos, 1995; [2,22]), the methodological approach employed by [22] seems the most reliable to date. Furthermore, the observer's bias is absent from this methodology.

Once we decide to use the data from [22] to validate or, at least, to support the selection of the best models generated by the distance sampling method, it is also necessary to analyze the data from the single morning surveys from 2008, 2012, and 2016 regarding the Azores bullfinch population dynamics. According to Ramos [25], who used data from recaptured ringed birds, the recruitment and annual mortality rates of the Azores bullfinch are similar, with a recruitment rate of 45 to 59% and a maximum annual mortality rate of 58%. The data from [22] corroborate [25]'s conclusions regarding the annual mortality rate, which was estimated at 62%. This information suggests that the Azores bullfinch population is prone to experiencing slight changes in its size every year. Therefore, we decided to accept the estimate presented by [19] (and the data from this study as well) since it seems more in accord with the estimation of 1608 individuals presented by [22]. Furthermore, if we

analyze the population trends provided by the population index estimation (Figure 5) and the population dynamics of the Azores bullfinch mentioned above, the population sizes estimated by [22] are not compatible with the study of [22]. In other words, it is unlikely that the Azores bullfinch population experienced a growth rate of more than 100% in a few years, from approximately 1500 to more than 3000 individuals in ten years. Additionally, the increase in population size noticed in 2006 was probably the consequence of a change in the methodology employed in the census, where the sampling effort was enlarged [3].

The Azores bullfinch has revealed a high adaptive capacity through physiologic and behavioral changes in response to profound human-induced modifications of its habitat [33]. This has allowed this species to persist under such disturbances and to make a population recovery in the last two decades. This plasticity is shown, for example, in the months with low food availability by the foraging on exotic plant species, such as *Clethra arborea* [2]. However, efforts to protect the remaining patches of Laurel Forest and actions to improve the environmental quality of habitats (such as controlling invasive plant species and introduced mammals) are crucial to ensuring the conservation of the Azores bullfinch. According to [34], who developed a spatial modeling framework to predict the Azores bullfinch responses related to native forest management, it is expected that in the next 25 years the population will increase by around 19% as a consequence of habitat management actions implemented since the beginning of the 2000s. This prediction agrees with the estimations from the single-morning surveys presented by [19] and by the present study since the increase in this species' population from 2008 to 2016 would be 14.3%, according to the data comparison produced in both studies. Therefore, we can conclude that the Azores bullfinch population is approximately 500 to 800 pairs, relatively stable, with an expected growth rate of around 15% each decade. The stability of the Azores bullfinch population and the expected growth depend on conservation efforts focused on habitat restoration. Additionally, as proposed by the last species action plan [35], monitoring Azores Bullfinch population size and breeding success are priority actions that must be implemented once a year. Further studies to update this species' ecology would also be of great interest in estimating its recovery and additional conservation needs.

**Author Contributions:** Conceptualization and methodology, S.T. and R.S.C.; writing—original draft preparation, T.d.M.M.C. and A.d.l.C.M.; writing—review and editing, A.G., S.T., R.S.C. and R.C. All authors have read and agreed to the published version of the manuscript.

**Funding:** This project had the financial support of the LIFE program of the European Commission and the Azores Government. Sergio Timóteo acknowledges funding from the FCT grant UID/BIA/04004/2020.

**Data Availability Statement:** The data presented in this study are available upon request from the corresponding authors.

**Acknowledgments:** The Priolo's Atlas is a citizen science project that is included in the Azores bullfinch monitoring program developed by the Portuguese Society for the Study of Birds since 2003. This project would not have been possible without the collaboration of all the volunteers that participated.

**Conflicts of Interest:** The authors declare no conflict of interest. The funders had no role in the design of the study; in the collection, analyses, or interpretation of data; in the writing of the manuscript; or in the decision to publish the results.

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
