# Peer review of "How Many Azores Bullfinches (Pyrrhula murina) Are There in the World? Case Study of a Threatened Species"

_diversity, doi:10.3390/d15050685_

Round 1

Reviewer 1 Report

Review for Diversity: Manuscript ID: diversity-2386475

Article: How many Azores bullfinches are there in the world? Case study of a threatened species

Comments to the Author

This paper deals with a subject that would be of interest to readers of Diversity and also deals with a subject that excites controversy within the biology and management conservation.

I really enjoyed reading the work. It seems very suitable for use next year in my university classes in the subject of Management and Conservation of Flora and Fauna. I think it summarises all the threats that a species can face, how to monitor it and, most positively, it presents a real case of how a re-evaluation according to the IUCN criteria can change a species from being critically endangered to not being critically endangered. In addition, I find it very interesting to highlight, on the one hand, how this species was hunted and brought to the brink of extinction because of the "damage" it caused to orange groves. And, on the other hand, to highlight the criteria used in this study to compare the data from different censuses.

In short, this is a nice paper. I think that is an interesting and useful contribution to the literature about bird census and management of a threatened species. In general the writing is clear. The arguments are clearly presented, the results are interesting and the interpretation of the results justified.

Minor comments:

-         Line 2. Shouldn't the scientific name be included?.

-         Line 57. Please insert the symbol “]" after the reference.

-         Line 63. Why was there no monitoring in those years?

-         Line 337-346. References 4, 5 and 6 have a high number of authors. In 4 and 6 it is marked as "et al.", but in reference 5 it is marked as "...." before the last author. Please use the same criteria.

-         Line 340. Enter the abbreviated name of the journal.

-         Line 389. Put the year in bold.

-         Line 392. Put the year in bold.

-         Line 398. Put the year in bold.

-         Line 408. The year is missing.

Sorry for my low level of written English.

Author Response

Dear Reviewer,

First of all, thanks for the comments addressed to the manuscript. You will find below the answers to each comment:

-         Line 2. Shouldn't the scientific name be included?

We agreed with this suggestion and the scientific name was included in the title.

-         Line 57. Please insert the symbol “]" after the reference.

Done.

-         Line 63. Why was there no monitoring in those years?

I didn't find information to explain the hiatus in the monitoring from 1997 to 2001. However, the monitoring restarted in 2002, and, in the next four years, more point counts and routes were added, especially in 2006 (see Ceia et al. 2011, page 479). These changes in the methodology were probably the most responsible for the "increasing" in the population number observed in 2006.

However, data in this study is analyzed from 2008, using two methods (annual monitoring and single morning surveys every four years) keeping the same sample efforts.

-         Line 337-346. References 4, 5 and 6 have a high number of authors. In 4 and 6 it is marked as "et al.", but in reference 5 it is marked as "...." before the last author. Please use the same criteria.

Reference 5 was formatted according to the journal format.

-         Line 340. Enter the abbreviated name of the journal.

Done.

-         Line 389. Put the year in bold.

Done.

-         Line 392. Put the year in bold.

In this case (book) the reference isn't in bold.

-         Line 398. Put the year in bold.

Done.

-         Line 408. The year is missing.

The reference was corrected following the journal format.

Reviewer 2 Report

see attached.

Author Response

Dear Reviewer,

Please, find attached the responses to your comments.

Best Regards,

Tarso Costa.

Round 2

Reviewer 2 Report

I think the authors have addressed well with my comments.

I have no further comments.